# Therapeutic efficacy of lenvatinib for patients with unresectable hepatocellular carcinoma based on the middle-term outcome

**Akira Fuchigami<sup></sup>, Yukinori Imai<sup></sup>, Yoshihito Uchida<sup></sup>, Hiroshi Uchiya[‡], Yohei Fujii[‡], Manabu Nakazawa[‡], Satsuki Ando[‡], Kayoko Sugawara[‡], Nobuaki Nakayama[‡], Tomoaki Tomiya[‡], Satoshi Mochida***

Gastroenterology & Hepatology, Faculty of Medicine, Saitama Medical University, Saitama, Japan

☯ These authors contributed equally to this work.
‡ These authors also contributed equally to this work.
* smochida@saitama-med.ac.jp

**Data Availability Statement:** All relevant data are within the manuscript.

**Funding:** Satoshi Mochida has received speaking fees or honoraria from AbbVie GK, Gilead Sciences

## Abstract

### Aim

This study sought to clarify the usefulness of lenvatinib for patients with unresectable hepatocellular carcinoma (HCC).

### Methods

The subjects were 69 patients with HCC receiving lenvatinib; the median age was 73 years, and 14 and 67 patients had been previously treated with regorafenib and/or sorafenib and therapies without molecular-targeted agents, respectively. Therapeutic efficacy was evaluated using contrast-enhanced CT images obtained 4–8 weeks after the start of lenvatinib and the middle-term outcome using Kaplan-Meier method.

### Results

The baseline Child-Pugh scores were 5, 6 and 7 in 31, 32 and 6 patients, respectively, and the modified albumin-bilirubin (mALBI) grades were 1, 2a and 2b in 20, 20 and 29 patients, respectively. The Barcelona Clinic Liver Cancer (BCLC) stages following downsizing after prior treatment were A, B and C in 17, 22 and 30 patients, respectively. The therapeutic efficacy was evaluated in 54 patients, and the percentages of patients achieving CR, PR, SD and PD were 3.7%, 44.4%, 37.0%, and 14.8%, respectively. The ALBI scores deteriorated significantly between 4 and 12 weeks after the start of therapy, compared with the baseline. The cumulative survival rates at 48 weeks were significantly higher among patients achieving CR/PR (95.5%) than among those showing no response (54.3%). Multivariate analyses revealed that the BCLC stages and the serum AFP levels were significantly associated with therapeutic efficacy, while the mALBI grade was associated with the middle-term outcome.

Inc., Otsuka Pharmaceutical Co., Ltd., Bristol Myers Squibb Co., Sumitomo Dainippon Pharma Co., ASKA Pharmaceutical Co., Ltd., Toray Medical Co. Ltd., Asahi Kasei Pharma Co., Kyowa Hakko Bio Co. Ltd., has received research grants from Gilead Sciences Inc., EA Pharma Co. Ltd., Janssen Pharmaceutical K.K., Kowa Co. Ltd., MSD K.K., AbbVie GK., Sumitomo Dainippon Pharma Co., Mochida Pharmaceutical Co. Ltd., Daiichi Sankyo Co. Ltd., Toray Medical Co. Ltd., Chugai Pharmaceutical Co. Ltd., SRL Inc., Japan Blood Products Organization. The funders had no role in study design, data collection and analysis, decision to publis h, or preparation of the manuscript. This does not alter our adherence to PLOS ONE policies on sharing data and materials.

**Competing interests:** Satoshi Mochida has received speaking fees or honoraria from AbbVie GK, Gilead Sciences Inc., Otsuka Pharmaceutical Co., Ltd., Bristol Myers Squibb Co., Sumitomo Dainippon Pharma Co., ASKA Pharmaceutical Co., Ltd., Toray Medical Co. Ltd., Asahi Kasei Pharma Co., Kyowa Hakko Bio Co. Ltd., has received research grants from Gilead Sciences Inc., EA Pharma Co. Ltd., Janssen Pharmaceutical K.K., Kowa Co. Ltd., MSD K.K., AbbVie GK., Sumitomo Dainippon Pharma Co., Mochida Pharmaceutical Co. Ltd., Daiichi Sankyo Co. Ltd., Toray Medical Co. Ltd., Chugai Pharmaceutical Co. Ltd., SRL Inc., Japan Blood Products Organization. The funders had no role in study design, data collection and analysis, decision to publis h, or preparation of the manuscript. This does not alter our adherence to PLOS ONE policies on sharing data and materials.

## Conclusions

A favorable middle-term outcome was obtained in patients with HCC receiving lenvatinib, especially in those manifesting grades 1/2a mALBI at baseline, despite the deterioration in ALBI scores during treatment.

## Introduction

Hepatocellular carcinoma (HCC) is the second leading cause of cancer death worldwide [1]. Molecular-targeted agents are recommended for the treatment of patients with well-preserved liver function (Child-Pugh classification of A) when HCC progression is diagnosed as Barcelona Clinic Liver Cancer (BCLC) stages B or C [2]. These agents were also shown to be effective for patients with BCLC stage B HCC who were refractory to transcatheter arterial chemoembolization (TACE) [3]. According to the REFLECT trial, a global multicenter randomized phase 3 trial, lenvatinib was shown to be not inferior to sorafenib when overall survival (OS) was used as the primary endpoint for patients with unresectable HCC [4]. In this trial, the objective response rate (ORR) was significantly higher among patients receiving lenvatinib (40.6%) than among those receiving sorafenib (12.4%) based on a masked independent imaging review using the modified Response Evaluation Criteria in Solid Tumors (mRECIST) [5]. In real-world practice, however, the OS rate (OSR) as well as the ORR and the relation between OSR and ORR are uncertain. Moreover, possible adverse events including a deterioration in liver function during lenvatinib administration remain to be elucidated. Thus, in the present paper, the middle-term outcomes of patients at 12 months after the initiation of lenvatinib were evaluated in relation to the short-term therapeutic efficacy assessed according to the mRECIST.

## Patients and methods

### Patients and study design

The subjects were 69 patients with HCC receiving lenvatinib at Saitama Medical University Hospital between March 2018 and June 2019. The demographic features and clinical characteristics of the patients were evaluated retrospectively. The study was approved by the Institutional Review Board of the Saitama Medical University Hospital (19080.01), and written informed consent for lenvatinib administration was obtained from all the patients at the initiation of treatment. Informed consent for the study was obtained in the form of opt-out consent on the website of Saitama Medical University Hospital (http://www.saitama-med.ac.jp/hospital/outline/irb_kouhou.html).

The extents of liver damage were assessed using the Child-Pugh classification and albumin-bilirubin (ALBI) score and the modified ALBI (mALBI) grades [6], while the extent of HCC progression was assessed according to the BCLC staging [2]. In patients receiving previous TACE, TACE refractoriness was determined according to the criteria proposed by the Japan Society of Hepatology (JSH) and the Liver Cancer Study Group of Japan [3, 7, 8].

### Therapy using lenvatinib and evaluation of therapeutic efficacy and adverse events

Lenvatinib (Eisai Co., Ltd, Tokyo, Japan) was administered once a day at a dose of 8 mg for patients with a body weight of less than 60 kg, and at a dose of 12 mg for those with a body

weight of 60 kg or more. The starting doses were reduced from 12 mg to 8 mg or from 8 mg to 4 mg in patients manifesting a Child-Pugh score of 7, and the doses were subsequently increased to the standard doses according to body weight if adverse events did not appear. In contrast, the doses were reduced or the therapies were discontinued when severe adverse events appeared.

The early therapeutic efficacy was evaluated using contrast-enhanced CT images obtained between 4 and 8 weeks after the initiation of lenvatinib therapy according to mRECIST, in which tumor response are assessed as follows; CR: the disappearance of any intratumor arterial enhancement in all target lesions, PR: at least a 30% decrease in the sum of diameters of viable (enhancement in the arterial phase) target lesions, PD: an increase of at least 20% in the sum of diameters of viable (enhancing) target lesions and SD: any cases that do not qualify for either PR or PD [5]. Adverse events were assessed according to the Common Terminology Criteria for Adverse Events (CTCAE), version 4.0, published by the National Cancer Institute [9].

## Statistical analysis

The therapeutic efficacies of lenvatinib were compared among patients manifesting different characteristics using the Fisher's exact test. A multivariate logistic regression analysis was performed to identify significant factors associated with early therapeutic efficacy. The Wilcoxon signed-rank test was used to compare liver function at baseline and that during the therapies. The time to tumor progression and the cumulative survival rates after the initiation of lenvatinib were calculated using the Kaplan-Meier method, followed by comparison using the log-rank test. The factors associated with survival rates were also analyzed using the Cox proportional hazard regression analysis. $P$ values of less than 0.05 were considered statistically significant.

## Results

### Demographic features and clinical characteristic of patients

The demographic features and clinical characteristics of 69 patients are shown in Table 1. The patients consisted of 45 men and 24 women with a median age of 73 years ranging from 51 to 86 years. The baseline Child-Pugh scores were 5, 6 and 7 in 31 (44.9%), 32 (46.4%) and 6 patients (8.7%), respectively, and the baseline mALBI grades were 1, 2a and 2b in 20 (29.0%), 20 (29.0%) and 29 patients (42.0%), respectively.

Among the 69 patients, 67 patients (97.1%) had undergone previous treatments for HCC that did not involve the use of molecular-targeted agents: liver resection had been performed in 16 patients, radiofrequency ablation (RFA) had been performed in 24 patients, and TACE and/or transcatheter arterial infusion chemotherapy (TAI) had been performed in 62 patients including 21 patients who were subsequently diagnosed as having TACE refractoriness. Moreover, 14 patients had received previous therapies with molecular-targeted agents: 10 patients had received sorafenib, and 4 patients had received sorafenib followed by regorafenib. HCC progression at the initiation of lenvatinib was classified as BCLC stages A, B and C in 17 (24.6%), 22 (31.9%) and 30 patients (43.5%), respectively. Among the 17 patients with BCLC stage A HCC at baseline, lenvatinib therapy was performed in 2 patients who had achieved HCC downsizing following partially successful TACE and/or TAI procedures done between 5 and 10 weeks before and in 14 patients who had not received RFA or surgical resection because of tumor location or the extent of liver damage and/or the wishes of the patients. One treatment-naïve patient received lenvatinib therapy because of an underlying lung disease that prohibited RFA, surgical resection and TACE using antineoplastic agents. Among the 22 patients with stage B HCC, lenvatinib therapy was performed following ineffective or partially

**Table 1. Demographic features and clinical characteristics of 69 patients with hepatocellular carcinoma receiving lenvatinib.**

| | | No. of patients (%) |
|---|---|---|
| Age: years old | medium (range) | 73 (51–86) |
| Sex | men : women | 45 (65) : 24 (35) |
| Etiology | HCV : HBV : alcohol : others | 37 (54) : 7 (10) : 13 (19) : 12 (17) |
| Child-Pugh score | 5 : 6 : 7 | 31 (45) : 32 (46) : 6 (9) |
| ALBI grade | 1: 2a : 2b | 20 (29) : 20 (29) : 29 (42) |
| BCLC stage | A : B : C | 17 (25) : 22 (32) : 30 (43) |
| Maximum tumor size (mm) | medium (range) | 28 (8–200) |
| Tumor multiplicity | 1 : 2 , 3 : 4–9 : ≥10 | 9 (13) : 26 (38) : 15 (22) : 19 (28) |
| Vp | 0 : 1 : 2 : 3 | 51 (74) : 7 (10) : 2 (3) : 9 (13) |
| Extrahepatic metastasis | absent : present | 56 (81) : 13 (19) |
| AFP (ng/mL) | <200 : ≥200 | 48 (70) : 21 (30) |
| Previous TACE/TAI therapy | none : once : twice : 3 times or more | 7 (10) : 11 (16) : 10 (14) : 41 (59) |
| TACE refractoriness | absent : present | 41 (66) : 21 (34) |
| Previous therapies using molecular-targeted agents | none: sorafenib : regorafenib following sorafenib | 55 (80) : 10 (14) : 4 (6) |

HCV: hepatitis C virus, HBV: hepatitis B virus, ALBI: albumin bilirubin, BCLC: Barcelona Clinic Liver Cancer, Vp; tumor thrombosis in the portal vein, AFP: alpha-fetoprotein, TACE: transcatheter arterial chemoembolization, TAI: transcatheter arterial infusion chemotherapy

successful TACE and/or TAI in 21 patients done between 5 and 64 weeks before. Another TACE-naïve patient received lenvatinib therapy for recurrence after hepatectomy done 49 weeks before. Among 30 patients with stage C HCC, lenvatinib therapy was performed following ineffective TACE and/TAI in 16 patients done between 3 and 27 weeks before. In all patients receiving the prior therapies for HCC, CT and/or MRI examinations were done from 3 to 81 weeks later, and lenvatinib administration was initiated within 17 weeks after the examinations. The extent of tumor thrombosis in the portal vein was classified as Vp1 (segmentary), Vp2 (secondary-order branch) and Vp3 (first-order branch) in 7, 2 and 9 patients, respectively. Extrahepatic metastasis was present in 13 patients (18.8%).

## Therapeutic efficacy of lenvatinib

Among the 69 patients, early therapeutic efficacy was only evaluated in 54 patients, since a contrast-enhanced CT examination had not been done because of impaired renal function and/or transfer to a regional hospital in 15 patients. The percentages of patients achieving CR, PR, SD and PD were 3.7%, 44.4%, 37.0% and 14.8%, respectively (Table 2). Thus, the ORR, which represents the total percentage of patients achieving a CR or PR, was 48.1%; the ORRs were 75.0%, 63.2% and 21.7% in patients with BCLC stage A, B and C, respectively, and the rates differed significantly depending on the extent of HCC progression ($P = 0.001$). Similarly, the ORRs were significantly higher in patients without extrahepatic metastasis than in those with metastasis (56.8% vs. 10.0%, $P = 0.012$) and in patients with serum AFP levels of <200 ng/mL than in those with levels ≥200 ng/mL (63.2% vs. 12.5%, $P = 0.001$), while it was not different between those with and those without portal vein tumor thrombosis (28.6% vs. 55.5%, $P = 0.124$). In contrast, early therapeutic efficacy did not differ depending on the ages of the

**Table 2. Early therapeutic efficacy of lenvatinib in patients with unresectable hepatocellular carcinoma (HCC) evaluated using contrast-enhanced Computed Tomography (CT) performed between 4 and 8 weeks after lenvatinib initiation according to the modified Response Evaluation Criteria in Solid Tumors (mRE-CIST) and factors associated with efficacy.**

| | | Univariate analysis | | | | | | Multivariate analysis | | |
| --- | --- | --- | --- | --- | --- | --- | --- | --- | --- | --- |
| | | Number of patients (%) | | | | | P values | Hazard ratio | 95% Confidence Interval | P values |
| | | Total | CR | PR | SD | PD | – | | | |
| Total | | 54 | 2 (3.7) | 24 (44.4) | 20 (37.0) | 8 (14.8) | – | | | |
| Age: years | < 70 | 17 | 0 | 8 (47.1) | 6 (35.3) | 3 (17.6) | 0.999 | | | |
| | ≥ 70 | 37 | 2 (5.4) | 16 (43.2) | 14 (37.8) | 5 (13.5) | | | | |
| Etiology | HCV | 29 | 0 | 13 (44.8) | 9 (31.0) | 7 (24.1) | 0.785 | | | |
| | Others | 25 | 2 (8.0) | 11 (44.0) | 11 (44.0) | 1 (4.0) | | | | |
| Child-Pugh score | 5 | 25 | 1 (4.0) | 14 (56.0) | 6 (24.0) | 4 (16.0) | 0.172 | | | |
| | 6, 7 | 29 | 1 (3.4) | 10 (34.5) | 14 (48.3) | 4 (13.8) | | | | |
| mALBI grade | 1 | 16 | 0 | 9 (56.3) | 5 (31.3) | 2 (12.5) | 0.554 | | | |
| | 2a, 2b | 38 | 2 (5.3) | 15 (39.5) | 15 (39.5) | 6 (15.8) | | | | |
| BCLC stages | A | 12 | 0 | 9 (75.0) | 2 (16.7) | 1 (8.3) | 0.001 | 6.64 | 1.70–25.87 | 0.006 |
| | B | 19 | 2 (10.5) | 10 (52.6) | 5 (26.3) | 2 (10.5) | | | | |
| | C | 23 | 0 | 5 (21.7) | 13 (56.5) | 5 (21.7) | | 1 | | |
| Vp | 0 | 40 | 2 (5.0) | 20 (50.0) | 11 (27.5) | 7 (17.5) | 0.124 | | | |
| | 1–3 | 14 | 0 | 4 28.6) | 9 (64.3) | 1 (7.1) | | | | |
| Extrahepatic metastasis | Absent | 44 | 2 (4.5) | 23 (52.3) | 15 (34.1) | 4 (9.1) | 0.012 | – | – | – |
| | Present | 10 | 0 | 1 (10.0) | 5 (50.0) | 4 (40.0) | | – | | |
| AFP: ng/mL | < 200 | 38 | 2 (5.3) | 22 (57.9) | 11 (28.9) | 3 (7.9) | 0.001 | 10.45 | 1.87–58.51 | 0.008 |
| | ≥ 200 | 16 | 0 | 2 (12.5) | 9 (56.3) | 5 (31.3) | | 1 | | |
| TACE, TAI | Naïve | 5 | 0 | 2 (40.0) | 1 (20.0) | 2 (40.0) | 0.999 | | | |
| | Experienced | 49 | 2 (4.1) | 22 (44.9) | 19 (38.8) | 6 (12.2) | | | | |
| | TACE refractoriness | 15 | 2 (13.3) | 8 (53.3) | 3 (20.0) | 2 (13.3) | – | | | |
| Molecular targeted agents | Naïve | 42 | 2 (4.8) | 20 (47.6) | 17 (40.5) | 3 (7.1) | 0.332 | | | |
| | Experienced | 12 | 0 | 4 (33.3) | 3 (25.0) | 5 (41.7) | | | | |

CR: complete response, PR: partial response, SD: stable disease, PD: progressive disease, mALBI: modified albumin bilirubin, BCLC: Barcelona Clinic Liver Cancer, Vp; tumor thrombosis in the portal vein, AFP: alpha-fetoprotein, TACE: transcatheter arterial chemoembolization, TAI: transcatheter arterial infusion chemotherapy

patients and their liver functions, such as the Child-Pugh scores and mALBI grades. The ORR was 49.0% in 49 patients receiving previous TACE and/or TAI and was 66.7% even in 15 patients diagnosed as having TACE refractoriness. In contrast, a CR was obtained in no patients, and the ORR was 33.3% in 12 patients who received previous treatment with regorafenib and/or sorafenib.

A multivariate analysis revealed that BCLC staging and the baseline serum AFP level were significant factors associated with early therapeutic efficacy, with odds ratios of 6.64 (A and B vs. C; $P = 0.006$) and 10.45 (<200 ng/mL vs. ≥200 ng/mL; $P = 0.008$) (Table 2), respectively.

The median time to tumor progression (TTP) in these 54 patients was 133 days (Fig 1A), and did not differ among patients with BCLC stages A, B and C HCC (164, 117 and 186 days, respectively (Fig 1B).

## Liver function in patients receiving lenvatinib

Liver function during lenvatinib administration was evaluated in 57 patients, since 5 patients were transferred to regional hospitals and treatment was discontinued because of adverse

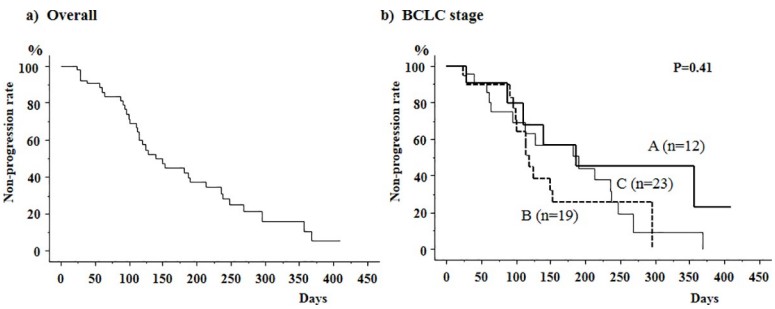

**Fig 1. Cumulative non-progression rates using the Kaplan-Meier method in 54 patients in whom early therapeutic efficacy was evaluated.** a) The median time to tumor progression (TTP) was 133 days. b) TTP did not differ depending on the Barcelona Clinic Liver Cancer (BCLC) stages. The median TTP were 164, 117 and 186 days in patients with stages A, B and C HCC, respectively.

events in 7 patients within 4 weeks after the initiation of lenvatinib. The median ALBI scores had deteriorated significantly at 4 weeks after the initiation of lenvatinib (-2.01, range -2.90 to -0.99), compared with the baseline scores (-2.42, range -3.16 to -1.40) ($P<0.01$), and the significant derangement persisted until 12 weeks; the median scores were -2.11 (-3.13 to -0.36) at 8 weeks and -2.13 (-2.99 to -0.39) at 12 weeks (Fig 2A). A similar deterioration was seen in patients manifesting mALBI grade 1 at baseline (Fig 2B) as well as those manifesting mALBI grades 2a and 2b (Fig 2C and 2D).

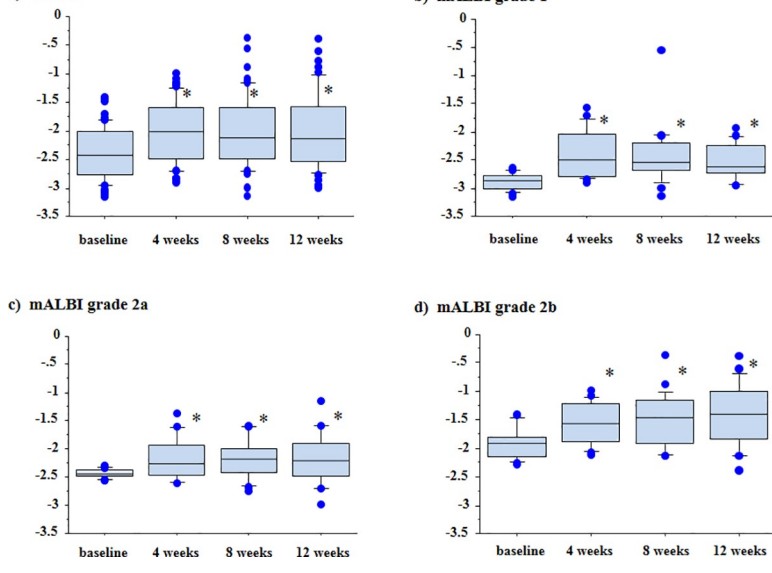

**Fig 2. Albumin-bilirubin (ALBI) scores in 57 patients with unresectable hepatocellular carcinoma (HCC) receiving lenvatinib for at least 4 weeks.** a) The median ALBI scores of patients had deteriorated significantly at 4 weeks (-2.01, range -2.90 to -0.99), 8 weeks (-2.11, range -3.13 to -0.36) and 12 weeks (-2.13, range -2.99 to -0.39) after the initiation of lenvatinib, compared with the baseline values (-2.42, range -3.16 to -1.40) (P<0.01). A similar deterioration was seen in patients manifesting modified ALBI (mALBI) grades 1, 2a and 2b at baseline. The median ALBI scores at baseline and at 4, 8 and 12 weeks after the initiation of lenvatinib were -2.87 (-3.16 to -2.65), -2.50 (-2.90 to -1.57), -2.53 (-3.13 to -0.54) and -2.60 (-2.96 to -1.93) in patients manifesting mALBI grade 1 (b), -2.44 (-2.56 to -2.30), -2.26 (-2.61 to -1.37), -2.19 (-2.76 to -1.58) and -2.21 (-2.99 to -1.14) in patients manifesting mALBI grade 2a (c), and -1.91 (-2.27 to -1.40), -1.56 (-2.11 to -0.99), -1.47 (-2.14 to -0.36) and -1.39 (-2.39 to -0.39) in patients manifesting mALBI grade 2b (d), respectively. *P<0.01 vs. baseline.

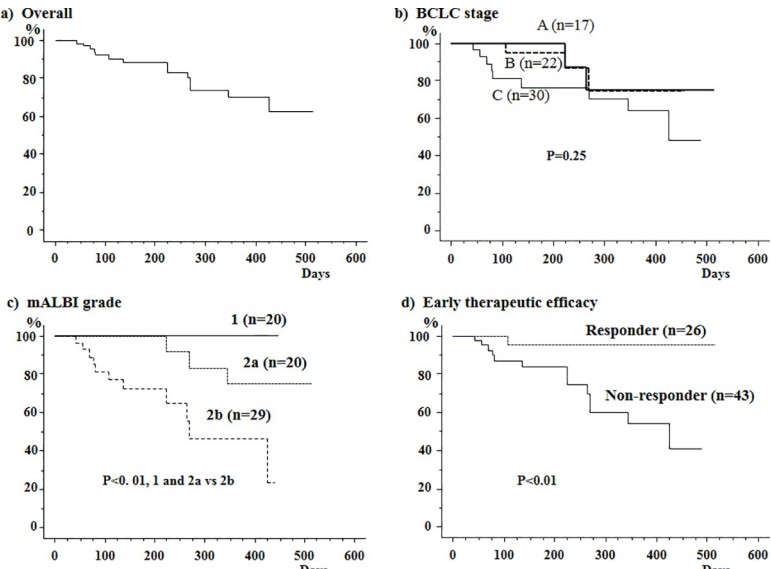

**Fig 3. Cumulative survival rates of 69 patients with unresectable hepatocellular carcinoma (HCC) receiving lenvatinib as evaluated using the Kaplan-Meier method.** a) The cumulative overall survival rates after the initiation of lenvatinib were 88.5% at 24 weeks and 73.9% at 48 weeks. b) The rates did not differ between patients with Barcelona Clinic Liver Cancer (BCLC) stages A and B HCC and those with stage C HCC. c) The rates in patients with modified albumin-bilirubin (mALBI) grade 1 and grade 2a at baseline were 90.9% at 48 weeks; these values were significantly higher than the rate in patients with mALBI grade 2b at baseline (46.6%) ($P<0.01$). d) The rates differed depending on the early therapeutic efficacies; the rates at 24 and 48 weeks were significantly higher in patients achieving CR and PR (95.5% and 95.5%, respectively) than in those showing no response (84.0% and 54.3%, respectively) ($P<0.01$).

## Outcome of patients receiving lenvatinib

The cumulative OSRs at 24 and 48 weeks after the initiation of lenvatinib were 88.5% and 73.9%, respectively (Fig 3A). The cumulative survival rates did not differ between patients with stages A and B HCC and those with stage C HCC (Fig 3B). Similarly, the rates did not differ between patients with and those without tumor thrombosis in the portal vein, those with and those without extrahepatic metastasis, and those with serum AFP levels <200 ng/mL and those with levels ≥200 ng/mL (Table 3). In contrast, the rates differed depending on liver function at baseline (Fig 3C); the rates at 48 weeks were 90.9% in patients manifesting mALBI grade 1 and grade 2a, and these values were significantly higher than the rate in patients manifesting mALBI grade 2b (46.6%) ($P<0.01$). Moreover, the outcomes of patients differed depending on the early therapeutic efficacy of lenvatinib as evaluated using contrast-enhanced CT between 4 and 8 weeks after the initiation of lenvatinib; the rates at 24 and 48 weeks were 95.5% and 95.5%, respectively, in patients achieving CR or PR, and both rates were significantly higher than the rates in those showing no response (84.0% and 54.3%, respectively; $P<0.01$) (Fig 3D).

The Cox proportional hazard regression method revealed that the mALBI grade was the only factor that was significantly associated with the cumulative OSRs of patients, with a hazard ratio of 0.16 (grades 1 and 2a vs. 2b; $P = 0.006$), while early therapeutic efficacy tended to be associated, with a hazard ratio of 0.14 (CR and PR vs. no response, $P = 0.062$) (Table 3).

## Adverse events

The adverse events seen during lenvatinib administration are shown in Table 4. Hypothyroidism was seen in 55 patients (81%), followed by proteinuria in 38 patients (55%), appetite loss

**Table 3. Cumulative survival rates of patients with unresectable hepatocellular carcinoma (HCC) receiving lenvatinib and factors associated with patient outcome.**

| | | Kaplan Meier method | | | Cox proportional hazard regression analysis | | |
| | | Cumulative Survival rates (%) | | P values | Hazard ratio | 95% Confidence Interval | P values |
| | | Total | 24 weeks | 48 weeks | | | | |
| Total | | 69 | 88.5 | 73.9 | – | | | |
| Age: years | < 70 | 23 | 84.6 | 76.2 | 0.79 | | | |
| | ≥ 70 | 46 | 90.2 | 72.7 | | | | |
| Etiology | HCV | 37 | 91.5 | 68.2 | 0.99 | | | |
| | Others | 32 | 84.5 | 84.5 | | | | |
| Child-Pugh score | 5 | 31 | 100 | 86.9 | 0.01 | – | – | – |
| | 6, 7 | 38 | 78.6 | 63.1 | | – | | |
| mALBI grade | 1, 2a | 40 | 100 | 90.9 | <0.01 | 0.16 | 0.04–0.59 | 0.006 |
| | 2b | 29 | 72.6 | 46.6 | | 1 | | |
| BCLC stages | A | 17 | 100 | 71.4 | 0.09 | | | |
| | B | 22 | 95.0 | 76.7 | | | | |
| | C | 30 | 76.3 | 70.4 | | | | |
| Vp | 0 | 50 | 93.3 | 76.8 | 0.41 | | | |
| | 1–3 | 19 | 74.2 | 64.9 | | | | |
| Extrahepatic metastasis | Absent | 56 | 89.9 | 71.3 | 0.37 | | | |
| | Present | 13 | 81.8 | 81.8 | | | | |
| AFP: ng/mL | < 200 | 48 | 93.2 | 74.0 | 0.10 | | | |
| | ≥ 200 | 21 | 73.9 | 73.9 | | | | |
| TACE, TAI | Naïve | 7 | 83.3 | 83.3 | 0.94 | | | |
| | Experienced | 62 | 89.1 | 73.3 | | | | |
| | TACE refractoriness | 21 | 90.5 | 70.4 | | | | |
| Molecular targeted agents | Naïve | 55 | 89.4 | 73.3 | 0.72 | | | |
| | Experienced | 14 | 84.6 | 74.0 | | | | |
| Early efficacy | CR, PR | 26 | 95.5 | 95.5 | <0.01 | 0.14 | 0.02–1.10 | 0.062 |
| | no response | 43 | 84.0 | 54.3 | | 1 | | |

mALBI: modified albumin bilirubin, BCLC: Barcelona Clinic Liver Cancer, Vp; tumor thrombosis in the portal vein, AFP: alpha-fetoprotein, TACE: transcatheter arterial chemoembolization, TAI: transcatheter arterial infusion chemotherapy, CR: complete response, PR: partial response, SD: stable disease, PD: progressive disease.

in 36 patients (52%), general fatigue in 31 patients (44%) and hand-foot syndrome in 28 patients (41%). Percentages of patients manifesting CTCAE grade 3 adverse events were 7% for thrombocytopenia and elevated aspartate aminotransferase, 5% for ascites, 4% for hepatic encephalopathy and increase of serum bilirubin and creatinine, 3% for appetite loss, decrease of serum albumin and proteinuria and 1% for general fatigue and hypothyroidism. Patients manifesting grade 4 adverse events were absent.

## Representative cases

A 78-year-old male patient with alcoholic liver cirrhosis was referred to our hospital for recurrence of HCC. He received hepatic left lateral segmentectomy 11 months ago, but multiple HCC recurrence occurred in the whole residual liver (Fig 4A). Contrast-enhanced CT images obtained 5 weeks after the initiation of lenvatinib therapy revealed decrease in diameters of enhancement lesions in the arterial phase, and the early therapeutic efficacy was diagnosed as PR (Fig 4B).

A 83-year-old male patient was referred to our hospital for the therapy of large HCC with a diameter of 130 mm. He received transcatheter arterial embolization (TAE) using

**Table 4. Adverse events seen during lenvatinib administration in patients with unresectable hepatocellular carcinoma (HCC).**

| Characteristics | Number of patients (%) | | | | |
|---|---|---|---|---|---|
| | Total Grading (CTCAE version 4) | | | | |
| | | 1 | 2 | 3 | 4 |
| Hand-foot syndrome | 28 (41%) | 24 (38%) | 4 (5%) | 0 | 0 |
| Rash | 5 (7%) | 2 (3%) | 3 (4%) | 0 | 0 |
| Diarrhea | 11 (15%) | 9 (13%) | 2 (2%) | 0 | 0 |
| Decreased appetite / Nausea | 36 (52%) | 14 (20%) | 20 (29%) | 2 (3%) | 0 |
| General fatigue | 31 (44%) | 20 (29%) | 10 (14%) | 1 (1%) | 0 |
| Ascites | 10 (14%) | 3 (4%) | 3 (4%) | 4 (5%) | 0 |
| Hypertension | 20 (28%) | 4 (5%) | 16 (23%) | 0 | 0 |
| Hepatic encephalopathy | 8 (11%) | 3 (4%) | 2 (3%) | 3 (4%) | 0 |
| Decreased neutrophil count | 11 (15%) | 3 (4%) | 8 (11%) | 0 | 0 |
| Anemia | 10 (14%) | 5 (7%) | 5 (7%) | 0 | 0 |
| Thrombocytopenia | 18 (25%) | 6 (8%) | 7 (10%) | 5 (7%) | 0 |
| Increased blood bilirubin | 18 (25%) | 4 (5%) | 11 (16%) | 3 (4%) | 0 |
| Elevated aspartate aminotransferase | 19 (27%) | 11 (16%) | 3 (4%) | 5 (7%) | 0 |
| Decreased serum albumin | 27 (39%) | 13 (20%) | 12 (17%) | 2 (3%) | 0 |
| Increased serum creatinine | 10 (14%) | 5 (7%) | 2 (3%) | 3 (4%) | 0 |
| Hypothyroidism | 55 (81%) | 20 (30%) | 34 (49%) | 1 (1%) | 0 |
| Proteinuria | 38 (55%) | 27 (40%) | 9(13%) | 2 (3%) | 0 |

microspheres for volume reduction of HCC. Contrast-enhanced CT images obtained 4 weeks after TAE revealed multiple enhanced lesions remained within the HCC nodule (Fig 5A). A CT examination performed 4 weeks following lenvatinib initiation revealed decrease of the

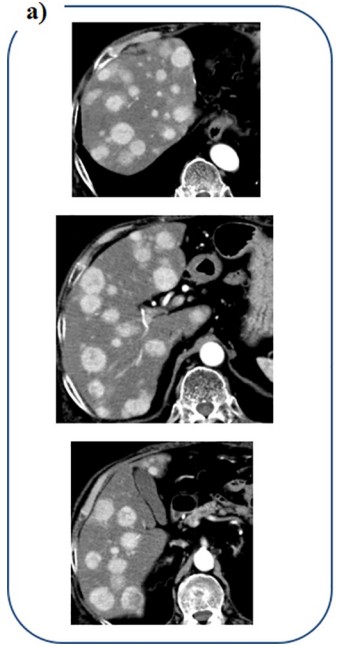
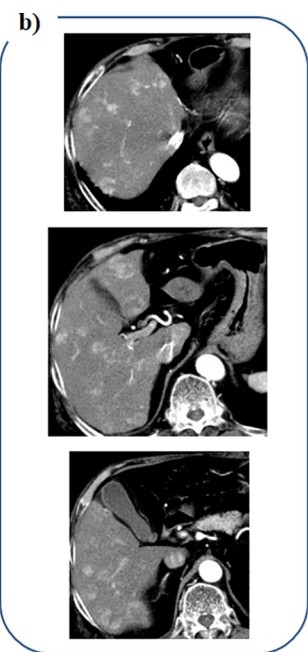

**Fig 4. Contrast- enhanced CT imaging in the arterial phase of a 78-year-old male patient with recurrent HCC.** a) Multiple HCC are seen in the whole liver before administration of lenvatinib. b) Enhancement lesions are decreased in diameters at 5 weeks after the initiation of lenvatinib.

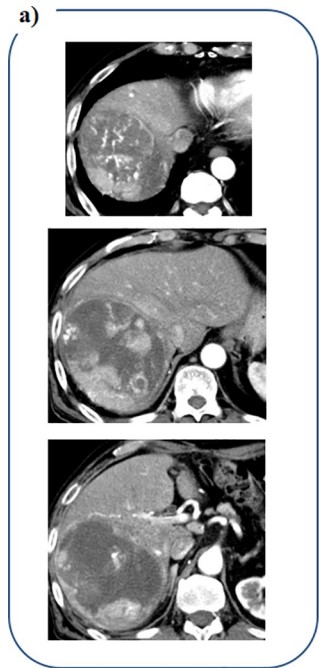
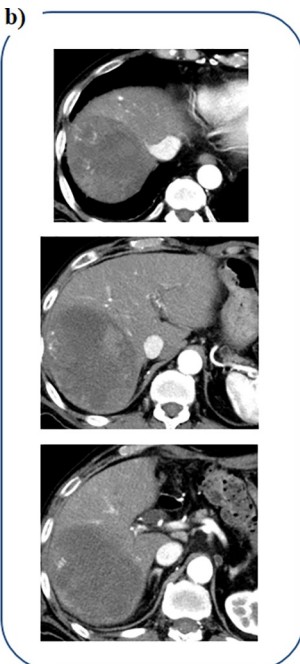

**Fig 5. Contrast-enhanced CT imaging in the arterial phase of a 83-year-old male patient with large HCC.** a) Multiple enhanced lesions are seen within the HCC nodule before administration of lenvatinib, despite TAE using microspheres was performed. b) The enhancement lesions are decreased in diameters at 4 weeks after the initiation of lenvatinib.

enhancement lesions in diameters (Fig 5B), and the early therapeutic efficacy was diagnosed as PR.

## Discussion

In the present study, the ORR in patients receiving lenvatinib as assessed using mRECIST based on contrast-enhanced CT imaging performed between 1 and 2 months after the initiation of treatment was 48.1%, and this value was similar to the value (40.6%) obtained in the REFLECT trial [4]. Similar results were obtained in real-world practice; Hiraoka *et al.* reported that based on observations made in a small cohort, a CR or PR was achieved in 40.7% of patients with unresectable HCC at 4 weeks after the initiation of lenvatinib [10]. In our cohort, the early therapeutic efficacies differed depending on the extent of HCC progression. The ORR was 63.2% in 19 patients with BCLC stage B HCC, while it was 21.7% in 23 patients with BCLC stage C HCC; the former value was equivalent to that of patients enrolled in the REFLECT trial, in which an ORR was achieved in 61.3% of the patients with BCLC stage B HCC [4]. An excellent therapeutic efficacy, as shown by an ORR of 75.0%, was also obtained in patients with BCLC stage A HCC receiving lenvatinib in whom those following downsizing of tumor volume by prior TACE and/or TAI were included, and an ORR of 66.7% was obtained even for patients with TACE refractoriness. These data suggest that lenvatinib may have an excellent early therapeutic efficacy in patients with intermediate stage HCC as a second-line therapy following TACE and/or TAI even if the prior therapies are ineffective.

According to the criteria proposed by the JSH and the Liver Cancer Study Group of Japan [3, 7, 8], TACE refractoriness is defined as the progression of HCC even after 2 or more consecutive TACE procedures in which the switching of antineoplastic agents and/or a reevaluation of the feeding arteries was adopted. Thus, in general, second-line therapies after TACE and/or

TAI are performed after at least two previous procedures that were ineffective. However, Hiraoka *et al.* reported that repeated TACE procedures can lead to a deterioration of liver function [11]; following each TACE procedure, the Child-Pugh scores had worsened to grade B in 9%-14% of patients who had been classified as grade A at baseline. We previously reported that the response rate of TACE using miriplatin was lower in patients receiving the procedure following unsuccessful TACE using antineoplastic agents other than miriplatin, compared with those receiving the procedure without prior TACE procedures [12]. Thus, in clinical practice, chemotherapy using lenvatinib is recommended to be performed without unnecessary TACE and/or TAI procedures in patients with intermediate stage HCC in whom TACE refractoriness is suspected based on their clinical features [13, 14], similar to chemotherapy using sorafenib [7]. This matter should be confirmed in the future studies. In the present study, a multivariate analysis revealed that serum AFP levels of less than 200 ng/mL as well as BCLC stages A and B were significant factors that were independently associated with the early therapeutic efficacy of lenvatinib. The significance of serum AFP levels has also been shown in treatment using sorafenib [15], since the tumor control rate is higher in patients with low serum AFP levels than in those with high serum AFP levels. Considering these observations, chemotherapy using lenvatinib merits consideration especially in patients with BCLC stage B HCC manifesting as low serum AFP levels prior to TACE and/or TAI procedures.

In the present study, the relation between objective response rates and survival outcome were evaluated in patients receiving lenvatinib, which were not clarified in the REFLECT trial and the previous study regarding the real-world practice [4, 10]. As shown in Table 3, the early therapeutic efficacy of lenvatinib therapy assessed using mRECIST was a significant factor associated with the cumulative survival rates of patients until 48 weeks after the initiation of lenvatinib in a univariate analysis and tended to also be associated with the middle-term outcome in the multivariate analysis. The EASL clinical practice guidelines recommend that the early therapeutic efficacies of loco-regional therapies for HCC be evaluated using mRECIST [2], and a meta-analysis revealed that the therapeutic efficacy according to mRECIST was a prognostic factor that was independently associated with the cumulative survival rates of patients receiving loco-regional therapies [16]. Moreover, early therapeutic efficacy according to mRECIST has been shown to be associated with the outcome of patients with unresectable HCC receiving sorafenib; the cumulative survival rates were higher in responders than in non-responders [17–20], and the SILIUS study, a prospective comparative study, revealed that the median survival duration of responders was 27.2 months, which was significantly longer than that in non-responders (8.9 months) [21]. These observations were in line with those made in the present study evaluating patients receiving lenvatinib.

In the present study, liver function evaluated using the mALBI grade was also crucial to determine the cumulative survival rates of patients with unresectable HCC receiving lenvatinib therapy. The cumulative survival rate was 100% in patients manifesting a mALBI grade 1 at baseline, and the impact of the mALBI grade on the middle-term outcome was similar to the early therapeutic efficacy of lenvatinib; the hazard ratios were 0.16 for mALBI grade (grades 1 and 2a vs. grade 2b) and 0.14 for early therapeutic efficacy (CR and PR vs. no response). Similar results were obtained in patients receiving chemotherapy using sorafenib for unresectable tumors [22, 23], regorafenib following sorafenib [24], and sequential therapies including lenvatinib [25]. These observations also prompted us to recommend that patients, in whom TACE refractoriness might be present receive chemotherapy using lenvatinib without receiving unnecessary TACE and/or TAI procedures. Of note, however, the liver function assessed using the ALBI scores deteriorated even in patients receiving lenvatinib during the 12 weeks of therapy, as shown in Fig 1. The impact of such an early deterioration in the ALBI scores on the long-term outcomes of patients should be investigated in the future.

As shown in Table 4, among various adverse events seen in patients receiving lenvatinib, hypothyroidism was the most frequent event, followed by proteinuria, appetite loss and general fatigue. Percentages of patients manifesting these events were higher than those shown in the REFLECT trial [4], while percentages of patients manifesting CTCAE grade 3 or more adverse events were in line with the trial.

The present study was done retrospectively in a single institute subjected for a small number cohort without a control group. Although early therapeutic efficacy of lenvatinib and the middle-term outcome of the patients were evaluated, effects of both outcome on the long-term outcome of patients were not clarified. Moreover, the heterogeneity of duration between the prior therapies and lenvatinib administration, number of the prior therapies and duration of CT and/or MRI examinations after the prior therapies may affect the middle-term outcome of patients. These limitations should be clarified prospectively in a multi-centric study subjected to large number cohort in the future.

In conclusion, chemotherapy using lenvatinib produced a high ORR in patients with BCLC stage B HCC or stage A HCC following prior TACE and/or TAI procedures, especially in those manifesting low serum AFP levels, and a favorable middle-term outcome was obtained in patients with mALBI grades 1 and 2a at baseline when they achieved a CR or PR according to mRECIST after between 4 and 8 weeks of therapy.

## Author Contributions

**Data curation:** Akira Fuchigami, Yukinori Imai, Yoshihito Uchida, Hiroshi Uchiya, Yohei Fujii, Manabu Nakazawa, Satsuki Ando, Kayoko Sugawara, Nobuaki Nakayama, Tomoaki Tomiya, Satoshi Mochida.

**Writing – original draft:** Akira Fuchigami, Yukinori Imai, Yoshihito Uchida, Hiroshi Uchiya, Satoshi Mochida.

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
