## [Decision Letter · Decision Letter 0]

13 Jan 2020

PONE-D-19-34992

Therapeutic Efficacy of Lenvatinib for Patients with Unresectable Hepatocellular Carcinoma Based on the Middle-Term Outcome

PLOS ONE

Dear Dr. Satoshi Mochida,

Thank you for submitting your manuscript to PLOS ONE. After careful consideration, we feel that it has merit but does not fully meet PLOS ONE’s publication criteria as it currently stands. Therefore, we invite you to submit a revised version of the manuscript that addresses the points raised during the review process.

We would appreciate receiving your revised manuscript within 60 days. To enhance the reproducibility of your results, we recommend that if applicable you deposit your laboratory protocols in protocols.io, where a protocol can be assigned its own identifier (DOI) such that it can be cited independently in the future. For instructions see: http://journals.plos.org/plosone/s/submission-guidelines#loc-laboratory-protocols

We look forward to receiving your revised manuscript.

Kind regards,

Gianfranco D. Alpini

Academic Editor

PLOS ONE

Journal Requirements:

2. Thank you for including your ethics statement: 'The study was approved by the Institutional Review Board of the Hospital (19080.01)'."  

a.Please amend your current ethics statement to include the full name of the ethics committee/institutional review board(s) that approved your specific study.  

b.Once you have amended this/these statement(s) in the Methods section of the manuscript, please add the same text to the “Ethics Statement” field of the submission form (via “Edit Submission”).

"Satoshi Mochida has received speaking fees or honoraria from AbbVie GK, Gilead Sciences Inc., Otsuka Pharmaceutical Co., Ltd., Bristol Myers Squibb Co., Sumitomo Dainippon Pharma Co., ASKA Pharmaceutical Co., Ltd., Toray Medical Co. Ltd., Asahi Kasei Pharma Co., Kyowa Hakko Bio Co. Ltd., has received research grants from Gilead Sciences Inc., EA Pharma Co. Ltd., Janssen Pharmaceutical K.K., Kowa Co. Ltd., MSD K.K., AbbVie GK., Sumitomo Dainippon Pharma Co., Mochida Pharmaceutical Co. Ltd., Daiichi Sankyo Co. Ltd., Toray Medical Co. Ltd., Chugai Pharmaceutical Co. Ltd., SRL Inc., Japan Blood Products Organization."            

4. Thank you for stating the following in the Financial Disclosure section:

"Satoshi Mochida has received speaking fees or honoraria from AbbVie GK, Gilead Sciences Inc., Otsuka Pharmaceutical Co., Ltd., Bristol Myers Squibb Co., Sumitomo Dainippon Pharma Co., ASKA Pharmaceutical Co., Ltd., Toray Medical Co. Ltd., Asahi Kasei Pharma Co., Kyowa Hakko Bio Co. Ltd., has received research grants from Gilead Sciences Inc., EA Pharma Co. Ltd., Janssen Pharmaceutical K.K., Kowa Co. Ltd., MSD K.K., AbbVie GK., Sumitomo Dainippon Pharma Co., Mochida Pharmaceutical Co. Ltd., Daiichi Sankyo Co. Ltd., Toray Medical Co. Ltd., Chugai Pharmaceutical Co. Ltd., SRL Inc., Japan Blood Products Organization."

We note that you received funding from a commercial source: Gilead Sciences Inc., EA Pharma Co. Ltd., Janssen Pharmaceutical K.K., Kowa Co. Ltd., MSD K.K., AbbVie GK., Sumitomo Dainippon Pharma Co., Mochida Pharmaceutical Co. Ltd., Daiichi Sankyo Co. Ltd., Toray Medical Co. Ltd., Chugai Pharmaceutical Co. Ltd., SRL Inc., Japan Blood Products Organization.

Reviewers' comments:

Reviewer's Responses to Questions

**Comments to the Author**

1. Is the manuscript technically sound, and do the data support the conclusions?

Reviewer #1: Yes

Reviewer #2: Yes

Reviewer #3: Partly

2. Has the statistical analysis been performed appropriately and rigorously? 

Reviewer #1: Yes

Reviewer #2: Yes

Reviewer #3: Yes

3. Have the authors made all data underlying the findings in their manuscript fully available?

Reviewer #1: Yes

Reviewer #2: Yes

Reviewer #3: Yes

4. Is the manuscript presented in an intelligible fashion and written in standard English?

Reviewer #1: Yes

Reviewer #2: Yes

Reviewer #3: Yes

5. Review Comments to the Author

Reviewer #1: This study aims to clarify the usefulness of lenvatinib for patients with unresectable HCC. 69 patients with HCC receiving lenvatinib were included. Therapeutic efficacy was evaluated using contrast-enhanced CT images obtained 4-8 weeks after the start of lenvatinib and the middle-term outcome using Kaplan-Meier method. The cumulative survival rates at 48 weeks were significantly higher among patients achieving CR/PR (95.5%). Multivariate analyses revealed that the BCLC stages and the serum AFP levels were significantly associated with therapeutic efficacy. Although this is an interesting report a real life management of HCC with the newly introduced drug lenvatinib some points should be clarified. 

Major comments:

1) Please dedicate a separate paragraph to safety and a table to adverse drug reaction/events... please discus this issue in comaprison with the registration trial.

2) It is not clear what is the timing of the starting of systemic therapy in patients receiving resection or locoregional therapy. Please specify. This seems particularly relevant because this affects the calculation of the survival in BCLC stages A and B. Indeed it was surprising that all BCLC stages have the same OSRs at 24 and 48 weeks. Arguably the follow up schedule before starting systemic therapy may affect the survival since an early detection of recidivism will be associated with a better survival with respect a recidivism diagnosed for symptoms. This should be discussed and whether high heterogeneity of follow up schedules will be exist this should be considered a limit of the study connected with the design. 

3) Thus, in clinical practice, chemotherapy using lenvatinib should be performed without unnecessary TACE and/or TAI procedures in patients with intermediate stage HCC in whom TACE refractoriness is suspected based on their clinical features [12, 13], similar to chemotherapy using sorafenib [7].

This is a strong statement not supported by evidence and not accepted by guidelines. 

Minor comments: Another missing information to be reported even in discussion is the number of TACE performed before starting lenvatinib.

Reviewer #2: This manuscript evaluated the therapeutic effect of lenvatinib for patients with unresectable HCC. The author found the early efficacy of lenvatinib was associated with BCLC stage and AFP levels. And liver function improved after 48 weeks. The cumulative survival rates only associated with mALBI grade.

Major points:

1. The adverse events should be evaluated among these patients

2. The CT imagine for pre and after the treatment for different cases should be included.

3. Besides the overall survival, it would be interesting to see the TTP number between different grade of HCC.

Minor point:

1. All the figures needs titles for each sub-figure for better interpreting.

Reviewer #3: In the present manuscript, Fuchigami A et colleagues investigated the therapeutic efficacy of Lenvatinib for patients with unresectable hepatocellular carcinoma (HCC). They included in this study 69 patients with HCC who were treated with lenvatinib. Authors individuated a favorable middle-term outcome in treated patients, despite the deterioration in ALBI scores during treatment.

Major concerns

- The manuscript lacks a control group; therefore, the results obtained by authors are almost descriptive. Moreover, the number of patients is limited. The authors should discuss this aspect and reduce their clinical claim in the discussion accordingly.

- It is not clear the novelty with respect to the cited Lancet paper (ref #4). Authors should better underscore the novelty of their manuscript.

- Authors should include the limitations of their study

- Table 1 should be improved. Headings should be included as well as percentages.

- mRECIST should be defined in methods

- Authors should be spelled out all abbreviations at their first use along the manuscript and also in abstract.

6. PLOS authors have the option to publish the peer review history of their article (what does this mean?). If published, this will include your full peer review and any attached files.

Reviewer #1: No

Reviewer #2: No

Reviewer #3: No

---

## [Author Response · Author response to Decision Letter 0]

19 Mar 2020

To Reviewer-1

1. Please dedicate a separate paragraph to safety and a table to adverse drug reactionevents. Please discus this issue in comaprison with the registration trial.

Descriptions regarding adverse events of lenvatinib were added to Patients & Methods (P4, L20-21), Results (P12, L1-L9) and were summarized in Table 4. In our cohort, hypothyroidism and proteinuria were seen more frequently in comparison with the registration trial, and this matter was described in Discussion (P16, L16-L20).

2. It is not clear what is the timing of the starting of systemic therapy in patients rceiving resection or locoregional therapy. Please specify.　This seems particularly　relevant because this affects the calculation of the survival in BCLC stages A and B.Indeed it was surpriing that all BCLC stages have the same OSRs at 24 and 48 weeks. Arguably the follow up schedule before starting systemic therapy may affect the survival since an early detection of recidivism will be associated with a better survival with respect a recidivism diagnosed for symptoms. This should be discussed and whether high heterogeneity of follow up schedules will be exist this should be considered a limit of the study connected with the design.

We added descriptions regarding the timing of lenvatinib initiation in patients with stages A, B and C HCC in Results (P5, L19-P6, L3) according to suggestion by the reviewer. The timing of CT and MRI examinations done following prior therapies before lenvatinib initiation was also shown in Results (P6 L3-L5). As pointed out by the reviewer, the heterogeneity of these timing may affect the outcome of patients receiving lenvatinib. These matters were described as the limitation of our study in Discussion (P16, L21-28).

3. Thus, in clinical practice, chemotherapy using lenvatinib should be performed without unnecessary TACE and/or TAI procedures in patients with intermediate stage HCC in whom TACE refractoriness is suspected based on their clinical features [12, 13], similar to chemotherapy using sorafenib [7].This is a strong statement not supported by evidence and not accepted by guidelines. 

 We agree with the comment by the reviewer. Thus, we changed the statement from ”should be” to “is recommended to be”, and added the description regarding further study in Discussion (P15, L4-7).

4. Another missing information to be reported even in discussion is the number of TACE performed before starting lenvatinib.

The description regarding the number of TACE and TAI procedures before lenvatinib administration was added to Table 1, and the heterogeneity of number was shown as limitation of the study which may affect the outcome of patients in Discussion (P16, L24-26) according to the suggestion by the reviewer. 

To Reviewer-2

1. The adverse events should be evaluated among these patients

Descriptions regarding adverse events of lenvatinib were added to Patients & Methods (P4, L20-21), Results (P12, L1-L9) and were summarized in Table 4. In our cohort, hypothyroidism and proteinuria were seen more frequently in comparison with the registration trial, and this matter was described in Discussion (P16, L16-L20).

2. The CT imagine for pre and after the treatment for different cases should be included

 We added two representative case reports in Results (P13 L1-P14, L1), and added Figure 4 and Figure 5. 

3. Besides the overall survival, it would be interesting to see the TTP number between grade of HCC.

We evaluated the time to tumor progression (TTP) according to the suggestion by the reviewer, and added description regarding these matters in Result (P8, L4- L6) and Figure 1.

4. All the figures needs titles for each sub-figure for better interpreting.

We added the title for each sub-figure in Figure 1, 2 and 3 according to the suggestion by the reviewer.

To Reviewer-3

1. The manuscript lacks a control group; therefore, the results obtained by authors are almost descriptive. Moreover, the number of patients is limited. The authors should discuss this aspect and reduce their clinical claim in the discussion accordingly.

We agree with the comment by the reviewer. Lack of control group and small number cohort were shown as the limitation of the present study in Discussion. (P16 L21-22).

2. It is not clear the novelty with respect to the cited Lancet paper (ref #4). Authors should better underscore the novelty of their manuscript.

The relation between objective response rates and survival outcome are uncertain in patients receiving lenvatinib both in REFLECT trial and previous reports regarding real-world practice. We added this description about the novelty of our manuscript in Discussion (P15, L15-17) according to the suggestion by the reviewer.

3. Authors should include the limitations of their study

We added the description regarding the limitations of our study according to the suggestion by the reviewer (P16, L21-28).

4. Table 1 should be improved. Headings should be included as well as percentages.

We added the heading and percentages of patients in Table 1 according to the suggestion by the reviewer.

5. mRECIST should be defined in methods

We added the description of mRECIST in Patients & Methods (P4, L15-21) according to the suggestion by the reviewer 

6. Authors should be spelled out all abbreviations at their first use along the manuscript and also in abstract. 

Spelled out all abbreviations were added to the revised Abstract according to the suggestion by the reviewer

---

## [Decision Letter · Decision Letter 1]

24 Mar 2020

Therapeutic Efficacy of Lenvatinib for Patients with Unresectable Hepatocellular Carcinoma Based on the Middle-Term Outcome

PONE-D-19-34992R1

Dear Dr. Satoshi Mochida,

We are pleased to inform you that your manuscript has been judged scientifically suitable for publication and will be formally accepted for publication once it complies with all outstanding technical requirements.

With kind regards,

Gianfranco D. Alpini

Academic Editor

PLOS ONE

Additional Editor Comments (optional):

Reviewers' comments:

Reviewer's Responses to Questions

**Comments to the Author**

1. If the authors have adequately addressed your comments raised in a previous round of review and you feel that this manuscript is now acceptable for publication, you may indicate that here to bypass the “Comments to the Author” section, enter your conflict of interest statement in the “Confidential to Editor” section, and submit your "Accept" recommendation.

Reviewer #1: All comments have been addressed

Reviewer #2: All comments have been addressed

Reviewer #3: All comments have been addressed

2. Is the manuscript technically sound, and do the data support the conclusions?

Reviewer #1: (No Response)

Reviewer #2: Yes

Reviewer #3: Yes

3. Has the statistical analysis been performed appropriately and rigorously? 

Reviewer #1: (No Response)

Reviewer #2: Yes

Reviewer #3: Yes

4. Have the authors made all data underlying the findings in their manuscript fully available?

Reviewer #1: (No Response)

Reviewer #2: (No Response)

Reviewer #3: Yes

5. Is the manuscript presented in an intelligible fashion and written in standard English?

Reviewer #1: (No Response)

Reviewer #2: Yes

Reviewer #3: Yes

6. Review Comments to the Author

Reviewer #1: (No Response)

Reviewer #2: The author addressed reviewers' comments pretty well. I have no further comments. The author has added suggested tables and figures.

Reviewer #3: (No Response)

7. PLOS authors have the option to publish the peer review history of their article (what does this mean?). If published, this will include your full peer review and any attached files.

Reviewer #1: No

Reviewer #2: Yes: Yuyan Han

Reviewer #3: No

---

## [Editor Report · Acceptance letter]

27 Mar 2020

PONE-D-19-34992R1 

Therapeutic Efficacy of Lenvatinib for Patients with Unresectable Hepatocellular Carcinoma Based on the Middle-Term Outcome 

Dear Dr. Mochida:

I am pleased to inform you that your manuscript has been deemed suitable for publication in PLOS ONE. Congratulations! Your manuscript is now with our production department. 

With kind regards,

on behalf of

Dr. Gianfranco D. Alpini 

Academic Editor

PLOS ONE